# Complex Repetitive Discharges: A Sign of Motor Axonal Reinnervation?

**DOI:** 10.3390/brainsci10060349

**Published:** 2020-06-05

**Authors:** Andreas Posa, Izabela Niśkiewicz, Alexander Emmer, Frank Hanisch, Malte E. Kornhuber

**Affiliations:** Department of Neurology, Martin-Luther-University Hospital Halle-Wittenberg, 06108 Halle, Germany; izabela-niskiewicz@o2.pl (I.N.); alexander.emmer@medizin.uni-halle.de (A.E.); hanisch.frank@gmx.de (F.H.); malte.kornhuber@medizin.uni-halle.de (M.E.K.)

**Keywords:** complex repetitive discharge (CRD), pathological spontaneous activity (PSA), electromyography (EMG), myo-axonal ephaptic re-excitation, axonal sprouting

## Abstract

Complex repetitive discharges (CRDs) are poorly understood phenomena in needle electromyography (EMG) recordings. The data presented here suggest that CRDs may mainly be a sign of motor unit reinnervation. EMG “video” data of 108 CRDs from neurogenic (ND, *n* = 39) and myogenic (MD, *n* = 14) disorders were retrospectively analyzed for cycle duration, potential-free time intervals, spike components (SC), maximum amplitudes, blockade, and increased jitter. CRD-SC in ND disorders (9.3 ± 7.8) outnumbered those in MD disorders (6.3 ± 6.2). The CRD cycle duration was correlated with SC and silent periods (*p* each < 0.000001). Blockade was observed in 36% and increased jitter in 27% of the CRDs. A higher number of CRD-SC in ND vs. MD fits the known differences in motor unit dimensions. Blockade and increased jitter are known features of diseased neuromuscular junctions, such as during reinnervation. The SC patterns of single CRD cycles resemble reinnervation potentials. Thus, CRDs may result from myo-axonal re-excitation in sprouting motor units. The purpose of this investigation was to better understand the circumstances under which CRDs may occur and eventually to contribute to the understanding of their pathogenesis.

## 1. Introduction

Pathological spontaneous activity in needle electromyography (EMG) investigations is frequently composed of single potential components, such as fibrillation potentials, positive sharp waves or high frequency discharges with or without variation in amplitude and intercomponent intervals. Not equally frequent is pathological spontaneous activity that consists of two or more spike components that cyclically discharge with more or less stable time intervals over time. The spikes, forming a cycle of such complex repetitive discharges (CRDs), have also been described as “the basic component” [1]. CRDs (Figure 1) may display characteristic noise behaviors that contribute to auditory CRD recognition. CRDs have been observed in different neuromuscular disorders, such as metabolic myopathies, muscular dystrophies, and motor neuron disorders, as well as in the lesions of nerve roots, nerve plexuses or peripheral nerves [2,3]. Their onset and offset are usually abrupt. The onset may be triggered by needle insertions [3,4,5].

CRDs have seldom been observed that were triggered by electrical motor nerve stimulation or by recruiting motor unit potentials [6]. In a single fiber EMG study, Trontelj and Stålberg (1983) suggested that CRDs are generated by ephaptic contacts between muscle fibers [2]. While CRDs have been reported to persist after nerve blockade by local anesthesia [7], pretreatment with botulinum toxin within the muscle may lead to their partial abolishment; however, in atypical cases with calf hypertrophy, CRDs are associated with an abundance of spontaneous muscle activity [8,9]. This finding may be taken as a possible hint for a presynaptic contribution to the occurrence of CRDs. Previously, blockade was reported in CRD recordings that would be in line with a myo-axonal ephaptic origin of CRDs [1,10]. Thus, presently, different hypotheses exist with respect to the pathogenesis of CRDs.

In the present investigation, a relatively large sample of 54 patients with CRDs in the EMG investigation was retrospectively analyzed. The purpose of this investigation was to analyze the features of CRDs in order to attempt to determine whether they are generated solely from muscle fibers or from contributions, e.g., from reinnervated nerves.

## 2. Materials and Methods

The data were investigated with the approval of the local ethics committee (Medical Faculty of the Martin-Luther-University of Halle-Wittenberg; No. 2017-108). Of 10,437 EMG investigations, in the period from 2001 to 2015, in the archives of the Department of Clinical Electrophysiology of Neurology at the Martin-Luther-University Hospital Halle (Saale), 1411 EMG recordings were available as “video” recordings with the EMG device (TOENNIES Multiliner and TOENNIES NeuroScreen) in the form of "-.wav" files. Storage in the video mode was routinely activated in case of rare and “interesting” EMG findings. The occurrence of a CRD constituted such rare and interesting findings. The 1411 video-EMG recordings were retrospectively evaluated for the presence of CRDs. A total of 108 CRDs were found in 54 patients. As the aim was to learn about the circumstances under which CRDs may occur, all 108 CRDs were analyzed, instead of taking only 1 CRD per patient. The CRDs were documented by 26-gauge concentric EMG disposable needle electrodes (S53156 or S53158, CareFusion, Madison, WI, USA). The filter settings were from 5 Hz to 5 kHz at an amplification of 50 µV per division and a time resolution of 10 ms per division (sample rate: 10 kHz). In a number of the CRDs, the maximum amplitude values could not be determined properly. As the CRDs were recorded with this high amplitude resolution, the upper and lower parts of CRD spike components could be lost due to saturation effects. As the EMG machine would delete the recordings after changing the amplification level, this loss in amplitude information had to be accepted.

The following data were recorded: age, sex, diagnosis, and the presence of a neurogenic or myogenic process. In one case with 2 CRDs, the latter assignment was not possible (see above).

The diagnoses of neurogenic and myogenic disorders were based on all available data, i.e., clinical, clinical chemical laboratory, electrophysiology, imaging, and biopsy studies, often including biochemical, histological, and genetic data.

The disease duration could not be unambiguously determined in the majority of cases. Therefore, this parameter was not taken into account. All analyses of the stored EMG data were conducted in a blinded fashion by experienced EMG investigators (AP, FH, MEK). For all data, consensus was obtained from all 3 investigators. The following parameters were included in the CRD-evaluation: affected muscle, recorded time interval of EMG “video” registration, CRD-duration within the video registration, time interval of each CRD cycle (Figure 2), type (fibrillation-like potentials and positive sharp wave-like potentials), total number of individual spike components within the cycle, rise time patterns of these individual discharges, duration of potential-free time intervals (Figure 2), changes in discharge patterns, blocking components, jittering components, and the maximum amplitude of the CRD-forming potential components. With respect to the rise times of the individual spikes within a given CRD cycle, the interest was whether these rise times were all similar or showed clearly discernable differences. Such differences were assumed when the ratio of the minimum divided by the maximum rise times of the spike components, measured in a given CRD cycle, were lower than or equal to 0.6. Thus, with a minimum rise time of, e.g., 0.2 ms and a maximum rise time of 0.7 ms, the ratio would be 0.29 and thus lower than 0.6.

In order to investigate the stability of the spike components in subsequent CRD cycles, we used peak triggering and a cascade display, as shown in Figure 3. Blocking was defined as the lack of an appearance of at least 1 in 10 of a certain CRD spike component.

A pathological increase in jitter was assumed when one of the spikes forming a CRD (i) showed a clearly visible variation in peak latency, relative to the other spike components, and (ii) showed a mean consecutive difference of at least 50 µs. As CRDs in most instances showed a highly constant discharge rate, the influence of the inter-discharge interval on the increase in jitter was not taken into consideration.

In fact, no clear reference values were established for this problem when concentric needle electrodes are used with conventional routine EMG filter settings. Such normative values were available for concentric needle electrode investigations only when special filter settings were used that differed considerably from those used in routine EMG investigations [11]. Changes in discharge patterns were analyzed. The duration of the CRD was recorded. When the CRD persisted to the end of the recording, the recorded time interval was taken as the duration. The recordings could be stored for up to a maximum duration of 1 min. When a CRD lasted longer, usually several sequences of the CRD were stored to assess its duration.

Statistical analyses were performed by Statistica Version 10 (StatSoft, Inc., Tulsa, OK, USA). Non-parametric test procedures were used. For the comparison of more than two samples, the Kruskal–Wallis test was used (a non-parametric analysis of variance). For the post-hoc analysis, the Mann–Whitney U test was used.

For the analysis of nominal variables, the Fisher–Freeman–Halton equivalent of the exact test used by Fisher was used. A Bonferroni correction was not performed for the post-hoc analysis in order not to overlook or discard too many actual contexts [12]. 

The data of the 1 patient with M. Pompe, who showed neurogenic as well as myogenic features in his EMG investigation, were not included in the statistical analysis. This patient showed two different CRDs, one "typical" discharge series and another CRD, at which time intervals with single discharges and those with complex discharge groups alternated (Figure 4). 

## 3. Results

There were 54 patients with 108 CRD recordings, age 56.4 ± 12.9 (28 females, age 59.4 ± 11.7 y, 26 males, age 53.2 ± 13.7 y; difference not statistically significant). In 39 patients, neurogenic disorders were present (age 59.1 ± 13.0 y; patient numbers in brackets): radicular syndromes (17), motor neuron diseases (13), polyneuropathies (6), and mononeuropathies (3). In 14 patients, myogenic disorders were present (48.9 ± 10.6 y): limb girdle muscular dystrophies (7), myositides (2), myotonic dystrophy type 2 (1), distal myopathy (1), and non-specified myopathies (3). The patients with myogenic disorders were significantly younger than those with neurogenic ones (*p* < 0.02). The additional patient with Pompe’s disease (45 y) was excluded from the analysis (see above). 

CRDs were identified in a diverse range of muscles (the number of neuropathy patients is before brackets; the number of myopathy patients is in brackets; when several CRDs were recorded in a muscle, this was taken as one; no patient showed CRDs in more than 1 muscle): biceps brachii 14 (3), erector spinae 11 (3), anterior tibial 7 (1), trapezius 2 (0), lateral vastus 1 (6), gastrocnemius 1 (1), brachioradialis 1 (0), iliopsoas 1 (0), and medial gluteus 1 (0). 

While in most muscles, only a single CRD was identified, there were muscles with 2 to 12 CRDs. CRD numbers per muscle did not differ significantly between neurogenic and myogenic disorders. A significant difference in the relative frequency of CRDs in a given muscle between neurogenic and myogenic disorders was only present in the lateral vastus (*p* < 0.005, Fisher’s exact test), a result that could also be due to the small number of patients.

The analysis of the CRD parameters is summarized in Table 1. The number of spike components per CRD cycle differed significantly between the myogenic and neurogenic processes. 

Furthermore, increased jitter was significantly more prevalent in the CRDs in neurogenic disorders than in myogenic ones. There was a positive correlation between the number of CRD spike constituents and the maximum amplitudes within a CRD cycle (Kendall’s tau = 0.29, *p* < 0.000013). The frequency distribution of the CRDs with different numbers of spike components per cycle is given in Figure 5. There was a highly significant positive correlation between the number of CRD components and the according cycle duration (Kendall’s tau = 0.38, *p* < 0.000001). The cycle duration by itself was highly positively correlated with the silent period as part of the CRD cycle (Kendall’s tau = 0.41, *p* < 0.000001).

## 4. Discussion

The frequency of the occurrence of CRDs was about 1 in 200 EMG investigations. This is similar to what was reported previously [4].

Previously, based on single fiber electromyography investigations, Trontelj and Stålberg (1983) proposed that CRDs may be generated ephaptically from one muscle fiber to the next [2]. Based on the data collected in the present investigation, albeit with a different methodology (a standard concentric needle electrode with routine filter settings), doubt about an ephaptic communication between muscle fibers as the only basis for CRDs was raised by the following findings: (a) the number of spike components per CRD cycle was significantly different in patients with myopathies and those with neurogenic disorders; (b) long silent periods were observed, separating subsequent spike components within a CRD cycle and between one cycle and the next; (c) single CRD spike components may show blockade or jitter; (d) the gain in the amplitude of CRD spikes may vary considerably within the same CRD cycle. Upon these findings, an alternative concept was outlined to explain the pathogenesis of CRDs, on the basis of one ephaptic contact between a muscle fiber and the motor unit by which it is innervated. This could happen if the nerve fiber is partly or completely demyelinated. The observation that CRDs can be abolished by botulinum toxin [8,9] is in line with the new concept of CRD generation. Botulinum toxin is taken up into the presynaptic terminals and reduces neuromuscular junction transmission [13].

The number of spike components was significantly larger in the CRDs of patients with neurogenic disorders, compared to those with myopathies. It is not clear how such a difference could be explained if the CRDs were generated ephaptically from one muscle fiber to the next, as inferred by Trontelj and Stålberg (1983) [2]. This difference fits the well-known differences between the numbers of muscle fibers per motor unit in muscles affected by neurogenic and myogenic disorders. The number of muscle fibers per motor unit in the fourth deep lumbricalis muscle of the healthy adult rat has been estimated to be around 70 [14]. In dystrophic myopathies, the number of muscle fibers per motor unit declined, while in neuropathies the respective number increased since the muscle fibers belonging to degenerating axons may be integrated into surviving motor units by way of axonal sprouting.

Signs of impaired neuromuscular transmission were reflected, e.g., by an increase in the jitter of subsequent spike components and by a blockade of spike components [15,16]. Besides, e.g., myasthenia gravis, an increase in jitter or blockade is often observed in parallel to the sprouting of reinnervating motor units [15,16]. In our sample, a blockade was detected in 36% of the CRDs (Figure 3). Increased jitter was present in 27 % of the recordings. An increase in jitter could develop abruptly while a CRD discharged and in one instance was involved in the ending of the CRD. Among the patients in our sample, there were no hints for other causes of a neuromuscular disorder, such as myasthenia gravis. Therefore, we take the observation of increased jitter or blockade in support of the view that motor axonal sprouting is involved in the generation of CRDs.

Demyelination may lead to delayed action potential propagation due to an impairment of the salutatory action potential propagation from one node of Ranvier to the next. In this respect, it is noticeable that CRD cycles could be separated or even interjected by more or less long potential-free ”silent” periods (e.g. Figure 1) that could last for more than 700 ms. Such long silent periods may be regarded as unlikely if the electrical coupling between muscle fibers is taken into consideration.

The CRDs were composed of spike components, with similar rise time patterns in the majority of the CRDs. However, the rise times of a proportion of the CRDs varied considerably and abruptly with minimum/maximum ratio values of ≤ 0.6 that could be also readily detected by eye (Figure 3; Table 1). The rise times of the myofiber action potentials detected in electromyography investigations depended on the distance between the myofiber and electrode tip. The rise times became shorter with a closer proximity between the myofiber and needle tip and vice versa. Differences in the rise time patterns of the neighboring potentials within the same CRD cycle may arise from the considerable spatial spread of the motor axon action potentials contributing to the CRD spikes, relative to the recording electrode. A larger area of potential spread could be the result of a reinnervation process. Reinnervation may also well explain the temporal dispersion of the spike components within a CRD cycle. If so, CRDs should be preferentially seen within a certain time period following a nerve or a nerve root lesion. This could be prospectively studied in parallel with routine EMG recordings, as in the present study and with the single fiber EMG technique. 

If CRDs were generated by the electrical coupling of a muscle fiber with an innervating axon, then this coupling could involve different sites of the axonal branches. If a muscle fiber was electrically coupled to a distal twig of the axon, few spike components would be expected to form the CRD, given that the action potential propagation in the proximal direction of the axon twig or branch is impossible, e.g., due to demyelinative conduction block. When the muscle fiber is coupled to a more proximal axon branch or to the principal axon, a larger number of spike components is expected to form the CRD. Strikingly, the numbers of the spike components that were present in CRDs were not evenly distributed. They peaked at the numbers of 2 to 3, 6 through 8, and 11 through 14 spike components (Figure 5). Such a distribution of the numbers of spike components per CRD would be well in line with the proposed electrical coupling of a muscle fiber at different sites of the branches of an axon. In this scenario, large numbers of spike components per CRD cycle would be in line with a relatively proximal motor axonal site of the ephapse. In the case of a distal location of the myo-axonal coupling, the re-excitation of the axonal twig by the excited muscle fiber is expected to be more rapid. Therefore, a short CRD cycle duration is expected. In the case of a more proximal ephapse site on the axonal tree, the re-excitation would take a longer time interval with a relatively longer CRD cycle duration. In line with this concept, a highly significant positive correlation was found between the number of CRD components and the according cycle duration. The outlined findings by no means exclude that, besides CRDs of a myo-axonal origin, there could also be CRDs due to ephapses solely between muscle fibers, as suggested by Trontelj and Stålberg (1983) [2]. The likelihood of such ephapses to be present declines rapidly with the number of the involved muscle fibers. Especially in the case of the CRDs in myopathy patients, such a decline in the frequency of CRDs with two to four potentials was visible (Figure 5). Therefore, these CRDs are special candidates for a myo-myal ephaptic origin. Thus, it could be that CRDs that arise from synapses exclusively between muscle fibers are more prevalent in the myopathy group of patients.

Previously, cyclically repetitive discharges of a motor unit were triggered by electrical motor nerve stimulation [6]. These axo-myo-axonal CRDs induced by an electrical axonal stimulus were short lived [6]. If the spontaneous CRD-trigger was located within the motor axon, the firing would be expected to be relatively inconstant, as known from the previously described SCMUSD (Spontaneous Continuous Motor Unit Single Discharges) [17]. Spontaneous CRDs may, however, discharge at a constant rate for prolonged periods of time. In the present investigation, a distinct CRD was recorded for more than 7 min. It might be that the sometimes long persistence of CRDs is due to a pacing muscle fiber. Indeed, we have recorded a CRD that alternated with a fibrillation potential (Figure 4). Eventually, the fibrillation potential was the pacemaker of the CRD that ceased after certain time intervals, e.g., due to ephaptic or motor axonal exhaustion. It could even be that part of the CRDs do not require a cyclic reinnervation of the pacing muscle fiber. The pacing fiber could then belong to a separate motor unit. This would enhance the likelihood of CRDs to occur.

When, during the axonal sprouting process, muscle fibers become reinnervated, after some time they usually stop generating pathological spontaneous activity. For instance, 8.5 ± 2.5 years after an accessory-to-suprascapular nerve transfer in 11 patients, the needle electromyography of the ipsilateral infraspinatus muscle showed chronic neurogenic changes but no pathological spontaneous activity [18].

CRDs are seen relatively often in ongoing reinnervation processes, such as in motor neuron diseases. When CRDs more or less represent a sign of axonal sprouting, as discussed above, they may be diagnostically meaningful in EMG investigations.

## 5. Conclusions

The purpose of the present study was to better understand the circumstances under which CRDs may occur and eventually to contribute to the understanding of their pathogenesis. A comprehensive and detailed analysis of CRDs was given in various neuromuscular disorders. The findings of the present study fit to the concept that CRDs are mainly due to the ephaptic excitation of parts of a demyelinated and reinnervating terminal motor axon sprout, e.g., by a spontaneously discharging muscle fiber in its close vicinity. Therefore, the majority of CRDs may be taken as a sign of an ongoing motor axonal reinnervation process.

## Figures and Tables

**Figure 1 brainsci-10-00349-f001:**
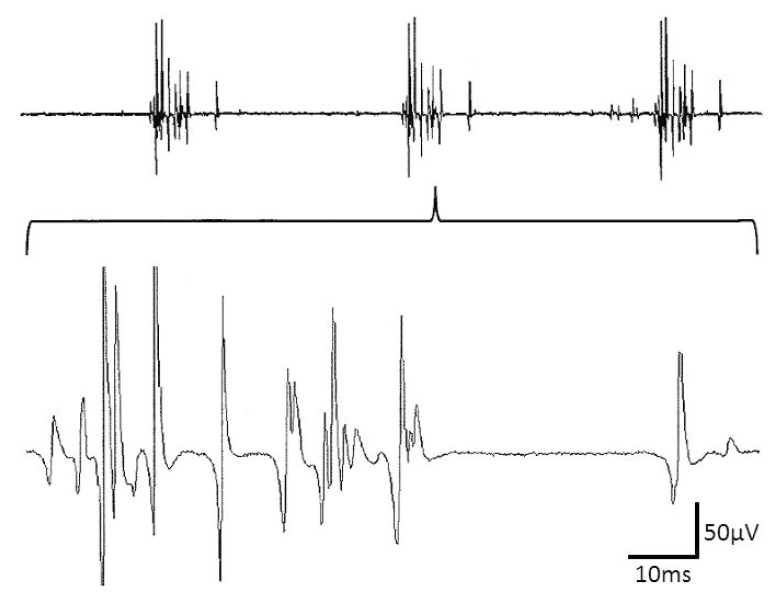
Complex repetitive discharges (CRD) with multiple spike components and a very large potential-free interval as part of the CRD cycle. Note that part of the spikes saturates.

**Figure 2 brainsci-10-00349-f002:**
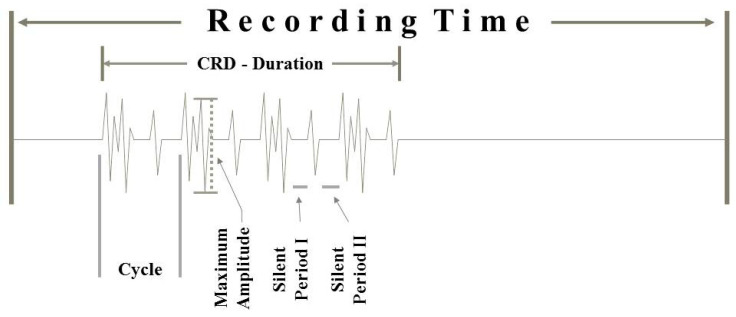
Schematic draft of a CRD recording showing the main CRD features that have been evaluated.

**Figure 3 brainsci-10-00349-f003:**
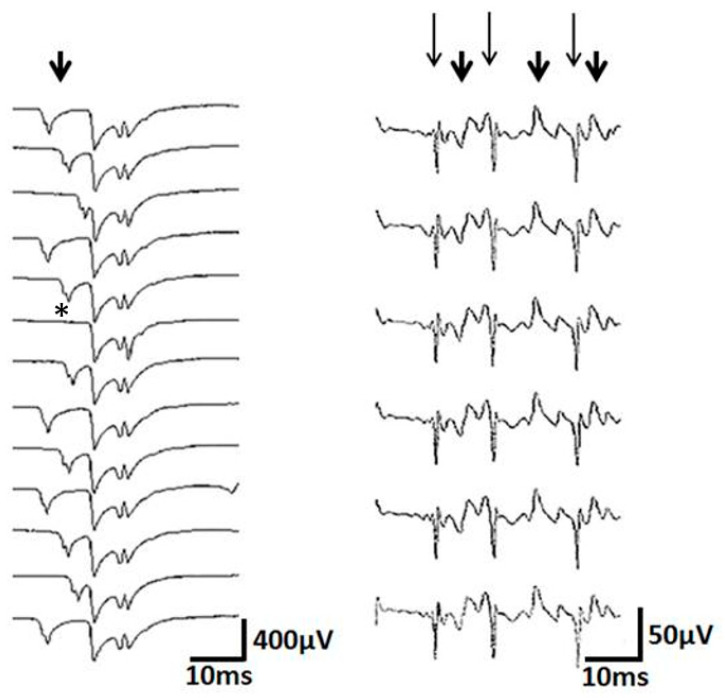
Left: note the blockade (*****) and jitter (arrow) of the spike components. Note also that the jittering part is composed of at least 2 potentials that jitter together. This may be due to a nerve twig taking part in the jitter. Right: subsequent complex repetitive discharges (CRD), shown in a cascade manner. Note that the high frequency spike components (long thin arrows) alternate with the low frequency spike components (short thick arrows). This contiguity of needle-near and needle-far spikes can hardly be explained by ephaptic transmission from one muscle fiber to the next.

**Figure 4 brainsci-10-00349-f004:**
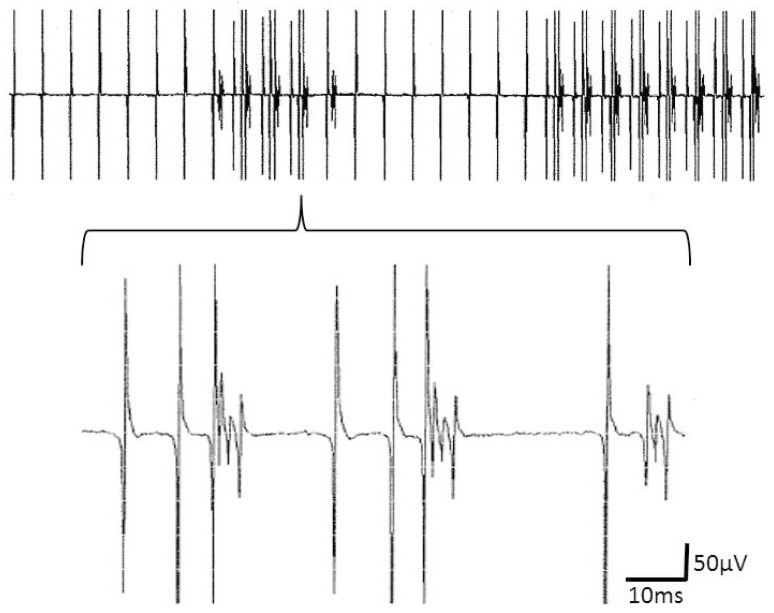
Upper part: complex repetitive discharges (CRD) that alternate with a fibrillation potential that takes part in the CRD, presumably as its pacemaker. The lower part shows the CRDs in an enlarged window.

**Figure 5 brainsci-10-00349-f005:**
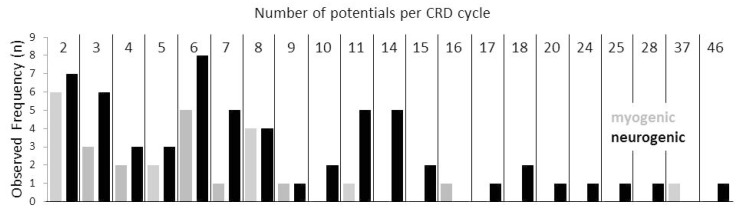
Relative complex repetitive discharge (CRD) frequencies are summarized in neurogenic (black columns) and myogenic disorders (grey columns) according to the numbers of single components forming a CRD cycle. Note that the observed relative frequencies are not evenly distributed but form peaks at single component numbers around 2/3, 6/7/8 and 11/14. Hypothetically, this type of distribution may correspond to certain sites of myo-axonal contacts within the axonal tree.

**Table 1 brainsci-10-00349-t001:** Summary of CRD results.

	Neurogenic Group	Myogenic Group	Both Groups
**Spike Components (n)**	9.3 ± 7.8 (2-46) *	6.3 ± 6.2 (2-37) *	8.4 ± 7.4 (2-46)
**“Fibrillation Potentials” (n)**	7.8 ± 6.7 (0-37) ^1^ **	4.3 ± 2.9 (0-12) ^2^ **	6.7 ± 6.0 (0-37) ^3^
**“Positive Sharp Waves” (n)**	1.5 ± 2.0 (0-9) ^1^	1.0 ± 1.7 (0-6) ^2^	1.4 ± 1.9 (0-9) ^3^
**Maximum Amplitude ^4^ (µV)**	247 ± 190 (10-1000)	181 ± 125 (20-675)	225 ± 175 (10-1000)
**Cycle Time Interval (ms)**	47.7 ± 48.5 (7-372)	76.3 ± 149.2 (8-748)	55.4 ± 92.5 (7-748)
**Potential-free Time Interval (ms)**	12.6 ± 25.2 (0-176)	45.2 ± 145.6 (0-725)	22.4 ± 84.4 (0-725)
**CRD Duration (s)**	27.3 ± 52.8 (1-420)	32.8 ± 103.2 (0.5-600)	28.8 ± 71.9 (0.5-600)
**Spatial CRD Spread**	13 of 73	6 of 33	19 of 106
**CRD Constancy**	51 of 73	25 of 33	76 of 106
**Differences in Spike Rise Times**	13 of 73	6 of 33	19 of 106
**Jitter**	26 of 71 ^1^ ***	1 of 30 ^2^ ***	27 of 101
**Blockade**	27 of 71 ^1^	9 of 30 ^2^	36 of 101

There was a total of 108 CRDs. Two recordings were not evaluated as a patient with Pompe disease could not be sorted into a neurogenic or myogenic group. The table is based on 106 CRDs: 73 from neurogenic and 33 from myogenic patients. In the subgroups of the data, further losses were present: (1) in 2 of the 73 CRDs, no differentiation between “fibrillation potentials” and “positive sharp waves” could be made due to the abundance of overlapping spike components; (2) in 3 of the 33 CRDs, no differentiation between “fibrillation potentials” and “positive sharp waves” could be made due to the abundance of overlapping spike components; (3) the loss of 2 CRDs from neurogenic patients and 3 from myogenic patients resulted in a sample of *n* = 101; (4) the maximum amplitude values may have been underestimated due to saturation effects, recorded at 50 µV/division (see text). * *p* < 0.05; ** *p* < 0.005; *** *p* < 0.0005.

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
