# Peer review of "Complex Repetitive Discharges: A Sign of Motor Axonal Reinnervation?"

_brainsci, 2020, doi:10.3390/brainsci10060349_

Round 1

Reviewer 1 Report

Re: Complex repetitive discharges, a sign of motor axonal reinnervation?

This study presented a retrospective analysis of complex repetitive discharges (CRDs) with the goal to explore the mechanism of this neuromuscular phenomenon. A number of parameters were calculated from CRDs and compared between neurogenic and myogenic conditions. The major findings included the neurogenic group had a significantly greater number of spike components, higher percentage of jitter and blockade. The number of potentials per cycle showed a sparse distribution in both groups.  The authors concluded that all these features could indicate CRD originating from the neuromuscular junction, where there could be ongoing reinnervation.

This paper provided an interesting explanation for the occurrence of CRDs, which may supplement the existing theory of ephaptic coupling between muscle fibers. While some of the results were supportive of this new theory, others may need further explanation or validation. In general, I feel the authors were biased towards their theory by neglecting the results in the myopathy group, which can possibly be explained by the existing myogenic theory. The authors should also provide more background information and in-depth discussion to improve the readability and comprehensiveness of this paper.

Specific comments are provided as follows:

  1. Lines 45-47: Botox treatment has led to the abolishment of CRDs – this could be strong evidence that there is a pre-synaptic contribution to the occurrence of CRDs. How is this related to your proposed theory? Why was it special in that atypical case? Was this evidence for or against the ephaptic theory? All these would need to be elaborated more clearly.
  2. Lines 45-48: It seems like blockade may or may not stop CRD, is there any difference in methodology between these studies? More information may be needed.
  3. Lines 53-54: As the proposed theory doesn’t sound mutually exclusive to the existing one based on the results and all previous evidence provided in the paper, I would recommend taking it as a supplement to the current theory (or make this statement clearer).
  4. Line 74: What does it mean?
  5. Line 91: Why 0.6?
  6. Figure 3 left: It would be better to use different arrows to show blockade and jitter, respectively
  7. Figure 5: The myogenic type distribution seems logarithmic (more number at the lower end), which could better be explained by the ephaptic coupling theory and fits the pathological changes of myopathy. The authors were inclined to ascribe these results to a single theory, which may bring in some biases. In line with comment 3, I would think the authors should at least acknowledge a potential difference in results between the two groups and apply their theory to explain both groups. If their theory can only explain the neurogenic group but not the myogenic group, it sounds a supplement instead of a replacement.
  8. Lines 196-201: In myopathy where the damage occurs only between some muscle membranes, the extent of the closed-circuit can be limited so that the number of CRD spike components is small; while in neuropathy demyelinated nerve terminals can result in a significantly larger area of activation (e.g. by motor unit). It still indicates there can be two different origins.
  9. Lines 236-239: More proximal coupling -> more fibers activated and longer re-excitation, and vice versa. Therefore, a correlation should be established between the number of spikes in a cycle and the silent interval between cycles to prove such a concept. It should not be between the number of spikes and the cycle duration, as this is always a strong positive correlation.
  10. A lot of statements in this article were based on the authors’ own opinions (a lot of “may”) or without proper citations (e.g. line 250), which makes their arguments less persuasive. I would recommend adding more citations wherever deemed necessary.

Author Response

REVISION 1

Article: Complex repetitive discharges, a sign of motor axonal reinnervation?

Review Report

Reviewer 1

This study presented a retrospective analysis of complex repetitive discharges (CRDs) with the goal to explore the mechanism of this neuromuscular phenomenon. A number of parameters were calculated from CRDs and compared between neurogenic and myogenic conditions. The major findings included the neurogenic group had a significantly greater number of spike components, higher percentage of jitter and blockade. The number of potentials per cycle showed a sparse distribution in both groups.  The authors concluded that all these features could indicate CRD originating from the neuromuscular junction, where there could be ongoing reinnervation.

This paper provided an interesting explanation for the occurrence of CRDs, which may supplement the existing theory of ephaptic coupling between muscle fibers. While some of the results were supportive of this new theory, others may need further explanation or validation. In general, I feel the authors were biased towards their theory by neglecting the results in the myopathy group, which can possibly be explained by the existing myogenic theory. The authors should also provide more background information and in-depth discussion to improve the readability and comprehensiveness of this paper.

Specific comments are provided as follows:

     Lines 45-47: Botox treatment has led to the abolishment of CRDs – this could be strong evidence that there is a pre-synaptic contribution to the occurrence of CRDs. How is this related to your proposed theory? Why was it special in that atypical case? Was this evidence for or against the ephaptic theory? All these would need to be elaborated more clearly.

Comment: We are grateful for the comment. Atypical in the 1 case of Costa et al. was the calf hypertrophy. Nix et al. described further 2 cases with CRD after radicular lesions with muscle pseudohypertrophy. Also in these 2 cases CRD were diminished after Botox treatment. We have added this reference. The 3 cases with diminution of CRDs after Botox treatment may be taken in favour of a presynaptic contribution to the occurrence of CRDs. We have described this point in further detail in the manuscript.

     Lines 45-48: It seems like blockade may or may not stop CRD, is there any difference in methodology between these studies? More information may be needed.

Comment: In lines 45-48 we did not state that blockade may or may not stop CRDs. Also  Partanen (Muscle Nerve 53: 508–512, 2016) did not mention that blockade may stop CRDs. There has been no principal difference in the methodology between Partanen’s study and ours. In lines 240-242 we mentioned that in 1 instance jitter was accompanied with ending of the CRD. It is unclear if and how the jitter led to this ending. 

     Lines 53-54: As the proposed theory doesn’t sound mutually exclusive to the existing one based on the results and all previous evidence provided in the paper, I would recommend taking it as a supplement to the current theory (or make this statement clearer).

Comment: We are grateful for this comment. We have changed our manuscript accordingly.

     Line 74: What does it mean?

Comment: This patient as diagnosed with a proximal myopathic myopathy. However, in the EMG results, there was also clear evidence for neurogenic features such as e.g. high amplitude motor unit potentials. Therefore, this patient was not assigned to the groups with neurogenic or myogenic patients.  

     Line 91: Why 0.6?

Comment: We have given an example now in the text. In our hands, the ratio of 0.6 discriminates well between CRDs that harbor potentials with similar risetimes and those with considerable differences in their risetimes. The 0.6-value as such cannot be founded theoretically, however.

     Figure 3 left: It would be better to use different arrows to show blockade and jitter, respectively

Comment: We are grateful for the suggestion. Figure and figure legend were changed accordingly.

     Figure 5: The myogenic type distribution seems logarithmic (more number at the lower end), which could better be explained by the ephaptic coupling theory and fits the pathological changes of myopathy. The authors were inclined to ascribe these results to a single theory, which may bring in some biases. In line with comment 3, I would think the authors should at least acknowledge a potential difference in results between the two groups and apply their theory to explain both groups. If their theory can only explain the neurogenic group but not the myogenic group, it sounds a supplement instead of a replacement.

Comment: We agree with the reviewer that part of the data may be explained by a myo-axonal synaptic origin while in other part of CRDs a synaptic origin between muscle fibers may be present. We have discussed this point as suggested.

     Lines 196-201: In myopathy where the damage occurs only between some muscle membranes, the extent of the closed-circuit can be limited so that the number of CRD spike components is small; while in neuropathy demyelinated nerve terminals can result in a significantly larger area of activation (e.g. by motor unit). It still indicates there can be two different origins.

Comment: This is a good notion. While we are not convinced that the difference in pathogenesis would account for the difference in spike components, we cannot rule out this possibility. The text was changed accordingly.

     Lines 236-239: More proximal coupling -> more fibers activated and longer re-excitation, and vice versa. Therefore, a correlation should be established between the number of spikes in a cycle and the silent interval between cycles to prove such a concept. It should not be between the number of spikes and the cycle duration, as this is always a strong positive correlation.

Comment: To be honest: There was no significant correlation (Kendall’s tau) between the number of spike potentials in the cycle and the 1st or 2nd silent period: p=0.17 for the 1st silent period and p=0.17 for the 2nd silent period. As far as we understand the matter, the silent period is mainly due to some demyelinated axon segment(s) mainly distal to the myo-axonal ephapse. There may or may not be major demyelination. The lack to demonstrate a significant correlation here does not mean anything in our opinion.

     A lot of statements in this article were based on the authors’ own opinions (a lot of “may”) or without proper citations (e.g. line 250), which makes their arguments less persuasive. I would recommend adding more citations wherever deemed necessary.

Comment: We are grateful for this comment. We have changed our manuscript accordingly.

Reviewer 2 Report

This report is an appropriate description of a rare EMG phenomenon, complex repetitive discharges (CRDs). It also conjectures the possible origin of CRDs, in a different way than earlier reports (Trontelj and Stålberg 1983, Partanen 2016). While new hypotheses are welcome they also need reasoning. The discussion about the problems with muscle fibre ephapse hypothesis presented by Trontelj ans Stålberg is convincing. The new hypothesis that the patohogenesis of CRDs involves one ephaptic contact between a muscle fiber and the motor unit by which it is innervated needs a reference and discussion. I could not find any experimental proof of myoneural ephapse, but this mechanism was suggested in an article of Magistris and Roth 1985 and it is still controversial. The influence of the proximal vs distal site of myoaxonal ephapse with respect to the motor unit as to the time interval and number of spike components needs more clarification. The action potential of a motor nerve twig travels to both directions from the point of ephapse spreading to all branches, and thus a motor unit potential should be observed irrespective of the distal or proximal site of myoneural ephapse.

Author Response

REVISION 1

Article: Complex repetitive discharges, a sign of motor axonal reinnervation?

Review Report

Reviewer 2

This report is an appropriate description of a rare EMG phenomenon, complex repetitive discharges (CRDs). It also conjectures the possible origin of CRDs, in a different way than earlier reports (Trontelj and Stålberg 1983, Partanen 2016). While new hypotheses are welcome they also need reasoning. The discussion about the problems with muscle fibre ephapse hypothesis presented by Trontelj ans Stålberg is convincing. The new hypothesis that the pathogenesis of CRDs involves one ephaptic contact between a muscle fiber and the motor unit by which it is innervated needs a reference and discussion. I could not find any experimental proof of myoneural ephapse, but this mechanism was suggested in an article of Magistris and Roth 1985 and it is still controversial.

Comment: We are grateful for the reviewer’s comment. In fact, myoneural synapses have been suggested by Roth and his group. The most convincing evidence for this phenomenon has been cited (Roth, G. Repetitive discharge due to self-ephaptic excitation of a motor unit. Electroencephalogr. Clin. Neurophysiol. 1994, 93, 1–6.). The phenomenon described by Roth was triggered by electrical nerve stimulation. In this aspect, it is different from the spontaneously occurring CRD. We discussed this point in further detail now.

The influence of the proximal vs distal site of myoaxonal ephapse with respect to the motor unit as to the time interval and number of spike components needs more clarification. The action potential of a motor nerve twig travels to both directions from the point of ephapse spreading to all branches, and thus a motor unit potential should be observed irrespective of the distal or proximal site of myoneural ephapse.

Comment: We agree with the reviewer that in principal myoaxonal ephaptic exation should eraise action potentials in the entire axonal tree regardless of where in this tree the ephapse is located. However, it cannot be taken for sure that action potentials may be safely transmitted over demyelinated axon segments. We have outlined this point in the discussion.